# Split_ Composite: A Radar Target Recognition Method on FFT Convolution Acceleration

**DOI:** 10.3390/s24144476

**Published:** 2024-07-11

**Authors:** Xuanchao Li, Yonghua He, Weigang Zhu, Wei Qu, Yonggang Li, Chenxuan Li, Bakun Zhu

**Affiliations:** 1Graduate School, Space Engineering University, Beijing 101416, China; lixuanchao22@163.com (X.L.); 15811314901@163.com (Y.L.); lccxmail@163.com (C.L.); zbk@hgd.edu.cn (B.Z.); 2Space Engineering University, Beijing 101416, China; yi_yun_hou@163.com (W.Z.); quweistar@163.com (W.Q.)

**Keywords:** ship target recognition, CNNs, fast Fourier transform, input block decomposition, composite zero-padding, inference speed

## Abstract

Synthetic Aperture Radar (SAR) is renowned for its all-weather and all-time imaging capabilities, making it invaluable for ship target recognition. Despite the advancements in deep learning models, the efficiency of Convolutional Neural Networks (CNNs) in the frequency domain is often constrained by memory limitations and the stringent real-time requirements of embedded systems. To surmount these obstacles, we introduce the Split_ Composite method, an innovative convolution acceleration technique grounded in Fast Fourier Transform (FFT). This method employs input block decomposition and a composite zero-padding approach to streamline memory bandwidth and computational complexity via optimized frequency-domain convolution and image reconstruction. By capitalizing on FFT’s inherent periodicity to augment frequency resolution, Split_ Composite facilitates weight sharing, curtailing both memory access and computational demands. Our experiments, conducted using the OpenSARShip-4 dataset, confirm that the Split_ Composite method upholds high recognition precision while markedly enhancing inference velocity, especially in the realm of large-scale data processing, thereby exhibiting exceptional scalability and efficiency. When juxtaposed with state-of-the-art convolution optimization technologies such as Winograd and TensorRT, Split_ Composite has demonstrated a significant lead in inference speed without compromising the precision of recognition.

## 1. Introduction

Synthetic Aperture Radar (SAR) is a sophisticated microwave imaging sensor that actively emits and receives signals, transforming the received energy into visual imagery [1]. With decades of technological evolution, SAR has advanced to produce high-resolution imaging data, offering superior image quality [2]. SAR stands out among other imaging modalities due to its distinct benefits. It maintains consistent imaging capabilities across various weather and lighting conditions, such as day, overcast, and night, establishing itself as a dependable tool for continuous, all-weather, and multi-environmental target detection and surveillance. Furthermore, SAR’s robust penetration ability allows it to see through clouds, fog, and light vegetation, providing insights into subsurface or concealed targets—a critical feature for military applications like aerial tactical defense and the detection of underground targets [3]. The field of automatic maritime vessel target recognition has garnered global attention. Utilizing SAR technology, precise identification and tracking of maritime vessels are feasible, yielding detailed vessel parameters such as dimensions, morphology, and dynamic information. These capabilities are of paramount importance for maritime traffic safety, ocean resource exploitation, and military intelligence gathering [4].

With the rapid advancement of deep learning techniques, its theories and methodologies have been extensively applied in the field of automatic target recognition in SAR imagery. CNNs, as the core of deep learning models, have garnered significant interest from researchers for their exceptional feature extraction and learning capabilities, particularly in the context of SAR ship recognition. However, the enhancement in model performance has led to increased complexity in network architecture and an expansion in the number of model parameters, posing challenges for real-time recognition of SAR ship targets. Especially on resource-constrained embedded and mobile devices, factors such as power consumption, memory constraints, and real-time requirements have become critical considerations [5].

In the quest for real-time SAR ship target recognition, the scientific community has been actively engaged in streamlining its system architecture to enhance efficiency and mitigate redundancy. Within CNN models, a plethora of parameters are deemed superfluous, with a minor fraction of the network shouldering the majority of computational tasks. CNN simplification can be approached from multiple angles. Quantization techniques are designed to diminish the bit-width of network parameters, consequently reducing memory requirements and conserving computational time and energy [6]. Parameter pruning strategies eliminate redundant network parameters, with pruning capable of occurring at various granularities, including weights, channels, and convolutional kernels [7]. Neural architecture search (NAS) represents methodologies for the manual or automated exploration of efficient network configurations [8]. In recent scholarly discourse, the pursuit of network structural simplicity has been actualized through hardware implementation methods, with a focus on leveraging a range of technologies to ensure the hardware feasibility of such networks. Parallelization, data and resource sharing, and pipelining exemplify the technological strategies underpinning the quest for hardware-optimized network structures [9].

The domain of CNNs has witnessed a surge of interest in frequency-domain implementations, which offer significant advantages for processing efficiency. Researchers have introduced innovative techniques leveraging the FFT to expedite convolution operations. Michael Mathieu et al. [10] pioneered the application of FFT-based convolution acceleration in CNNs with an algorithm that hastens both training and inference during the three-pass propagation process, optimized for scenarios where filter dimensions approximate input feature maps. Abtahi et al. [11] advanced computational efficiency by employing the overlap-and-add convolution (OaAconv), effectively reducing complexity from *O(N*^2^*log*_2_*N)* to *O(N*^2^*log*_2_*K)*, particularly beneficial in CNN contexts where input sizes N are substantially large. Lin et al. [12] introduced tFFT, a Fourier-domain decomposition strategy that facilitates block transformations in the Fourier domain, accelerating convolutions involving small kernels and extensive inputs. Furthermore, Lin et al. [13] developed a novel framework integrating a unique Fourier-domain propagation process with innovative activation functions and subsampling techniques, streamlining the training and inference processes of CNNs in the Fourier domain. Hu’s [14] FFT1d-Conv stands out for its application of a one-dimensional FFT algorithm, sidestepping the computational overhead of higher-dimensional FFTs and enhancing efficiency. Sunny et al. [15] unveiled Spectral-Blaze, an FFT-based CNN accelerator that innovates through internal patch parallelization during the Hadamard product phase, optimizing MAC unit utilization and establishing a uniform reuse pattern across patches of input feature maps. This approach not only simplifies the tiling process but also enables selective patch element processing for on-chip memory efficiency, effectively mitigating computational and energy bottlenecks inherent in spatial domain acceleration strategies.

While existing methods have demonstrated the potential to expedite the training and inference processes of CNNs, they are not exempt from the computational overhead associated with Fourier-domain transformations, which require FFT and inverse FFT operations between convolutional layers. This frequent domain conversion is bandwidth-intensive and detracts from the desired efficiency of frequency-domain convolutions [16]. The primary sources of this inefficiency are twofold: firstly, the necessity for complex weight dimensions to align with those of the input complex mapping, necessitating the expansion of traditional convolutional weights through zero-padding to match the input size, followed by a Fast Fourier Transform [17]. This process escalates the computational complexity from *O(k*^2^*)* to *O(N*^2^*)*, with N and K representing the dimensions of the input map and kernel in traditional convolution, respectively. Given that the kernel size is typically much smaller than the input map, the associated overhead is notably substantial. Secondly, in contrast to traditional convolutions that facilitate weight sharing, the direct implementation of frequency-domain convolution lacks the capacity for shared complex weights, leading to increased memory access costs that counteract computational efficiency gains. These factors are detrimental to the precise and efficient execution of real-time Synthetic Aperture Radar (SAR) ship target missions. In response to these challenges, we introduce the Split_ Composite method, an FFT-based convolution acceleration technique specifically designed to expedite the real-time identification of ship targets in SAR imagery on resource-constrained embedded platforms.

Our key contributions are as follows:We introduce an input block decomposition technique that segments SAR ship images into smaller blocks, performing convolution operations on each in the frequency domain, followed by image reconstruction through an overlap-and-add approach. This methodology significantly reduces memory bandwidth consumption associated with FFT and inverse FFT operations while concurrently diminishing the computational complexity of the algorithm;Considering the absence of shareable complex weights and the high memory access overhead in the implementation of frequency-domain convolutions, which may undermine the gains in computational efficiency, we propose a hybrid zero-padding method. This approach leverages the periodic nature of the FFT by embedding zeros within the original weights, enhancing frequency resolution, and generating a shareable pattern of complex weights. This not only reduces the number of memory access operations but also lowers the complexity of arithmetic computations;We have constructed an integrated convolution acceleration scheme termed Split_ Composite, which initially segments the input SAR ship images into smaller blocks. Subsequently, it employs a hybrid zero-padding strategy to process the input blocks and convolutional kernels, followed by frequency-domain transformation and all operations associated with the FFT. Experimental results demonstrate that this comprehensive approach further enhances the performance of CNNs, rendering them more accurate and efficient in addressing the real-time identification of SAR ship targets.

The structure of this paper is delineated as follows. Section 2 elaborates on the FFT-based convolution acceleration method proposed in this study, detailing the computational steps involved in frequency-domain convolution, the methodology of input block decomposition, and the composite zero-padding technique, complemented by theoretical underpinnings and analytical discussions. Section 3 details the experimental substantiation of our proposed method, utilizing the OpenSARShip-4 dataset, which includes an overview of the experimental setup and data, an assessment of the input block decomposition outcomes, an evaluation of the composite zero-padding results, and a comparative analysis of performance against alternative convolution optimization techniques. Section 4 concludes the paper by encapsulating the principal contributions, delineating the merits and prospective applications of our FFT-based convolution acceleration approach, and offering an interpretive synthesis of the experimental findings.

## 2. Methods

### 2.1. Frequency-Domain Convolution

The process of frequency-domain convolution, as outlined in Figure 1, is composed of three essential stages: Initially, both the original convolutional weights *w* and the input feature maps *x* are translated into the frequency domain through the application of the FFT. Subsequently, an element-wise multiplication of these transformed quantities is executed in the frequency domain. Finally, the Inverse Fast Fourier Transform (IFFT) is applied to yield the resultant convolution output. The conventional convolution operation in the spatial domain is effectively replaced by this sequence of operations in the frequency domain, as detailed in the subsequent equations.

Let us consider a sample input $x\in R^{m_{0}\times m_{0}}$ and a sample filter weight $w\in R^{k\times k}$, where $k<m_{0}$. The convolution operation can be mathematically represented as follows:(1)$$y\left[n\right]=x\left[n\right]*w\left[n\right]=\sum_{m}w[m]\cdot x[n-m],$$
where ∗ is the convolution operation. *w* denotes the convolutional kernel (weights), *x* represents the input feature map, y signifies the convolutional output, and *n* and *m* are the indices of the sequences involved.

Initially, it is necessary to apply the Discrete Fourier Transform (DFT) to both the convolutional kernel *w* and the input feature map *x*:
(2)$$W[k]=\sum_{n}w[n]e^{-j\frac{2\pi }{N}kn},$$
(3)$$X[k]=\sum_{n}x[n]e^{-j\frac{2\pi }{N}kn},$$

Herein, *W*[*k*] and *X*[*k*] represent the DFT results of *w*[*n*] and *x*[*n*], respectively, where *N* denotes the sequence length and *k* is the frequency index.

According to the convolution theorem, the convolution in the time domain is equivalent to multiplication in the frequency domain:(4)$$y_{FFT}=W[k]\times X[k],$$

In this context, $y_{FFT}$ represents the product of the convolution kernel and the input feature map in the frequency domain.

Ultimately, the product obtained must be subjected to the Inverse Discrete Fourier Transform (IDFT) to retrieve the final convolution result:(5)$$y=\sum_{k}y_{FFT}\cdot e^{j\frac{2\pi }{N}kn}=\sum_{k}W[k]\cdot X[k]\cdot e^{j\frac{2\pi }{N}kn},$$

Here, y denotes the resultant convolution output. *j* is an imaginary unit. In this work, m=n=N. j and k are indexed from 1 to N.

The prerequisite for the FFT transformation involves the augmentation of the convolutional weights *w* to dimensions equivalent to those of the input image through a zero-padding operation, as exemplified in Figure 1. This procedure is essential because the subsequent element-wise multiplication in the frequency domain necessitates that the complex weights and activations be of the same size. Figure 2a delineates the zero-padding process for an individual input channel of the weight matrix *w*. Specifically, the original *k* × *k* weight matrix is expanded to *m*_0_ × *m*_0_ by padding zeros around its boundary. Although prior research [18] has proposed postponing the padding to after the FFT transformation, as indicated in Figure 2b, this approach leads to considerable computational overhead due to the interpolation process in the frequency domain.

### 2.2. Input Block Decomposition

The input block decomposition technique, as detailed in this section, involves dividing an *N* × *N* input array into smaller blocks, each of size ⌈*N*^2^*/n*^2^⌉, designed to match the dimensions of the convolution kernel, *n* × *n*. This method is exemplified in Figure 1, which outlines the process within a CNN architecture featuring three input channels and a single output channel. (1) The initial step involves segmenting the SAR image, sized *N* × *N*, into blocks that can vary in size according to hardware specifications and other operational constraints. (2) To reconcile the size mismatch between the decomposed blocks and the convolution kernel, zero-padding is applied to the convolution blocks pre-FFT to ensure uniformity in size. The blocks and the kernel are then concurrently transformed into the frequency domain through the FFT process, where they undergo element-wise convolution. (3) The resultant convolutional outputs are overlapped by *n* − 1 instances and aggregated. The aggregated results are subsequently transformed back into the spatial domain using the IFFT, yielding an output equivalent to that of the conventional spatial convolution method. The regions of overlap in the resultant output *y* are distinctly marked to illustrate the integration of the overlapped convolutional responses. (See Figure 3).

The input block decomposition technique facilitates the efficient computation of each convolution in the frequency domain, with the computational complexity for the two-dimensional FFT for each block being *O*(*n*^2^log_2_*n*). Consequently, the aggregate complexity for processing the entire input array and the convolution kernel is determined by the product of the number of blocks and the per-block convolution complexity, resulting in *O*(*N*^2^log_2_*n*). Table 1 presents a comparative analysis of the computational complexities associated with various convolutional methods. Herein, SpaceConv denotes the conventional spatial domain convolution, FFTConv signifies convolution executed in the frequency domain via Hadamard product without prior block decomposition of the input image, and SplitConv indicates the convolution approach that follows input image block decomposition, employing the overlap-and-add technique for efficient computation of smaller convolutions in the frequency domain. This comparison elucidates the efficiency gains achieved through the proposed input block decomposition technique, particularly in the context of large-scale input arrays and convolution kernels.

In the prevalent scenario within CNN architectures where the input dimension *N* substantially exceeds the kernel size *n*, SplitConv achieves a substantial reduction in computational complexity compared to SpaceConv and FFTConv. Specifically, the complexity is reduced by a factor of *n*^2^/log_2_(*n*) for SpaceConv and log_2_(*N*)/log_2_(*n*) for FFTConv. To illustrate with a concrete example, for a 128 × 128 input array—a common size in CNNs—paired with a 5 × 5 convolutional kernel, the computational complexity is markedly different across the methods: SpaceConv exhibits a complexity of *O*(128^2^ × 25), FFTConv has a complexity of *O*(128^2^ × 8), and SplitConv demonstrates a notably lower complexity of *O*(128^2^ × 2.3). This comparison underscores the efficiency of the SplitConv approach, particularly for large-scale inputs typical in CNN operations.

### 2.3. Composite Zero-Padding

Equation (5) illustrates the intrinsic periodicity of the FFT, where an *m* × *n* matrix is populated with *n* discrete sample points along each dimension, repeating periodically with a period of *n*. This work takes full advantage of the FFT’s periodic attribute, notably to augment the frequency resolution and establish a pattern of recurring weights. In line with the findings reported in the literature [19], the FFT can enhance its frequency resolution by incorporating zeros into the initial data input. We have adopted this zero-embedding technique to amplify the frequency resolution of the FFT, which in turn engenders repetitive weights. However, the sole introduction of zeros is not always sufficient to achieve the desired padding dimension *m*_0_ due to the discrepancies in input image sizes that exist between various layers within a CNN architecture and even among different neural network models. This observation underscores the need for a more dynamic approach to zero-padding to cater to the diverse dimensional specifications inherent in CNNs.

#### 2.3.1. Composite Zero-Padding Approach

This manuscript integrates the operations of zero-stuffing and zero-padding to offer a versatile padding approach tailored for the target dimensions, as depicted in Figure 2c. In the zero-stuffing phase, subsequent to each row/column of the original weights (demarcated by the orange region), *p* − 1 rows/columns of zeros are appended, with *p* being an integer no less than 2. During the subsequent zero-padding phase, zeros are added to the periphery of the weights until they attain dimensions of *m*_1_ × *m*_1_, where *m*_1_ is calculated as ⌈*m*_0_/*p*⌉⋅*p*, employing the ceiling function ⌈⋅⌉ and considering *m*_0_ as the initial size of the input mapping. Utilizing the composite zero-padding strategy, the FFT is capable of generating *p*^2^ replicated complex weight blocks, each of size (*m*_1_/*p*) × (*m*_1_/*p*). An exemplification is provided in Figure 2c, which displays four replications of complex weights, aligning with *p* = 2 and thus *p*^2^ = 4. This technique is pivotal for enhancing the frequency resolution and creating a pattern of recurring weights that can be leveraged across the FFT process.

The current study introduces two principal distinctions from the conventional zero-padding approach shown in Figure 2a: First, the incorporation of zeros into the original weight tensor, which is instrumental in augmenting the sampling density during the FFT transformation. Second, the padding of the weights to dimensions of *m*_1_ × *m*_1_, which approximate *m*_0_ × *m*_0_, produces replicated complex weights. This methodology, in contrast to the scenario in Figure 2b, circumvents the considerable computational expense inherent in frequency-domain padding.

#### 2.3.2. Theoretical Derivation

This section will delve into a theoretical exposition to establish that the FFT transformation, leveraging the composite zero-padding method, is capable of generating *p*^2^ sets of replicated complex weight values for an *m*_1_ × *m*_1_ input weight matrix. This derivation underscores the efficiency and innovation of the composite zero-padding technique in enhancing the FFT’s capacity to handle complex weight matrices in neural network applications.

Following the composite zero-padding operation (denoted as *w*_hp_), the FFT computation for the weights is performed as illustrated in Equation (6).
(6)$$Y_{u^{\prime },v^{\prime }}^{\prime }=FFT(w_{\mathrm{hp}})=\sum_{j^{\prime }=1}^{m^{\prime }}\sum_{l^{\prime }=1}^{n^{\prime }}W_{j^{\prime },l^{\prime }}^{\prime }\omega_{m^{\prime }}^{j^{\prime }(u^{\prime }-1)}\omega_{n^{\prime }}^{l^{\prime }(v^{\prime }-1)},$$

The symbol *W*′ refers to the weight matrix that has undergone the composite zero-padding procedure. Here, *n*′ is equal to *m*′, which is set to *m*_1_, indicating the dimensions of the transformed weight matrix. Additionally, the indices *u*′, *v*′, *j*′, and *l*′ are constrained to the interval [1, *m*_1_], representing the valid range of indices for accessing elements within the *m*_1_ × *m*_1_ matrix. This specification is essential for accurately referencing elements during the subsequent computational processes within the FFT-based convolutional framework.

We decompose the complex weight matrix *Y*′ into *p*^2^ distinct segments, with each segment possessing dimensions of (*m*_1_/*p*) × (*m*_1_/*p*). Here, *m*_1_ is an integer multiple of *p*. Our objective is to illustrate that the complex weights across these segments are identical, a proposition that can be mathematically expressed as follows: (7)$$\left\{\begin{array}{c} Y_{u_{0}^{\prime }+g.n_{1}/p,v_{0}^{\prime }+h.n_{1}/p}^{\prime }=Y_{u_{0}^{\prime },v_{0}^{\prime }}^{\prime };1\leq u_{0}^{\prime }\leq n_{1}/p;\\ 1\leq v_{0}^{\prime }\leq n_{1}/p;u_{0}^{\prime }+g.n_{1}/p\leq n_{1};h.n_{1}/p\leq n_{1} \end{array}\right.,$$

Here, *g* and *h* represent temporal integer parameters, each constrained to values greater than or equal to one (*g* ≥ 1 and *h* ≥ 1). Due to the application of a hybrid zero-padding pattern, the weight matrix *W*′ is articulated in the subsequent formulation:(8)$$W_{j^{\prime },l^{\prime }}^{\prime }=\left\{\begin{array}{c} W_{j^{\prime }/p,l^{\prime }/p};j^{\prime },l^{\prime }is\;the\;multiple\;of\;p\\ 0;Otherwise \end{array}\right.,$$

The symbols *W*′ and *W* represent the composite zero-padding weight matrix (*w*_hp_) and the original weight matrix (*w*), respectively. Utilizing Equation (8), we can effectively remove the null components from *W*′, which were initially introduced in Equation (6).
(9)$$Y_{u^{\prime },v^{\prime }}^{\prime }=\sum_{j^{\prime }=1}^{m^{\prime }}\sum_{l^{\prime }=1}^{n^{\prime }}W_{j^{\prime }/p,l^{\prime }/p}\omega_{m^{\prime }}^{j^{\prime }(u^{\prime }-1)}\omega_{n^{\prime }}^{l^{\prime }(v^{\prime }-1)},$$

Assuming *j* = *j*′/*p* and *l* = *l*′/*p*, Equation (9) can be reformulated as follows:(10)$$Y_{u^{\prime },v^{\prime }}^{\prime }=\sum_{j^{\prime }=1}^{m^{\prime }}\sum_{l^{\prime }=1}^{n^{\prime }}W_{j,l}\omega_{m^{\prime }}^{jp(u^{\prime }-1)}\omega_{n^{\prime }}^{lp(v^{\prime }-1)},$$

Referring to Equation (10), we can provide a detailed exposition of *Y*′ as defined in Equation (7), which can be articulated as follows:(11)$$Y_{u_{0}^{\prime }+g.n_{1}/p,v_{0}^{\prime }+h.n_{1}/p}^{\prime }=\sum_{j^{\prime }=1}^{m^{\prime }}\sum_{l^{\prime }=1}^{n^{\prime }}W_{j,l}\omega_{m^{\prime }}^{jp(u_{0}^{\prime }-1)}\omega_{n^{\prime }}^{lp(v_{0}^{\prime }-1)}=Y_{u_{0}^{\prime },v_{0}^{\prime }}^{\prime },$$

Considering the periodic nature of the functions *w_m_*_′_ and *w_n_*_′_, characterized by a period of *m*_1_, they can be expressed with the following formula:(12)$$\omega_{m^{\prime }}^{jg.n_{1}}=1;\;\omega_{n^{\prime }}^{l.g.n_{1}}=1,$$

In this case, Equation (11) can be reformulated as follows:(13)$$Y_{u_{0}^{\prime }+g.n_{1}/p,v_{0}^{\prime }+h.n_{1}/p}^{\prime }=\sum_{j^{\prime }=1}^{m^{\prime }}\sum_{l^{\prime }=1}^{n^{\prime }}W_{j,l}\omega_{m^{\prime }}^{jp(u_{0}^{\prime }-1)}\omega_{n^{\prime }}^{lp(v_{0}^{\prime }-1)}=Y_{u_{0}^{\prime },v_{0}^{\prime }}^{\prime },$$

The sequence of equations from (8) to (13) delineates the process by which *p*^2^ sets of complex weight values, as presented in Equation (7), are replicated through the application of the composite zero-padding technique in conjunction with the FFT. Specifically, Equation (13) reveals the condition under which these replicated values manifest: when *m*_1_ is a multiple of *p*.

### 2.4. Overview of the Split_ Composite Methodology

The method advanced in this manuscript, designated as Split_ Composite, is executed through a series of critical steps, as outlined in Figure 4. (1) The process initiates with the partitioning of the *N* × *N* SAR image into blocks of arbitrary size, tailored to meet hardware specifications and other constraints, with a presumed dimension of *m*_0_ × *m*_0_ for this study. (2) Following segmentation, the spatial weights and input mappings are expanded to dimensions of *m*_1_, preceding the application of the FFT. The spatial weights undergo padding in accordance with a composite zero-padding strategy. Should *m*_1_ surpass *m*_0_, the input mappings are padded with zeros to synchronize the dimensions for frequency-domain convolution. Subsequently, convolution in the frequency domain is conducted utilizing a set of shareable complex weights, as depicted in Figure 5. The complex weights ($Y_{w_{1}}^{\prime }$) exemplified in Figure 5 are repurposed across four distinct blocks of complex input mappings (*B*_0_, *B*_1_, *B*_2_, and *B*_3_), signifying the capability to apply a singular set of weights across multiple input blocks, thereby optimizing computational efficiency and curtailing computational resource consumption. This approach significantly reduces memory access by capitalizing on the replication of complex weights. (3) Concluding the process, the convolutional outcomes are overlapped *m*_1_ times, equivalent to the span of the convolution kernel, aggregated, and reverted to the spatial domain through the IFFT, yielding an output analogous to that of conventional spatial convolution, with the overlapping regions in the output *y* distinctly emphasized.

Contrasting with the traditional FFT-based convolution method illustrated in Figure 2a, the method we introduce enhances efficiency by enabling the reuse of complex weights across various blocks within the input maps. This innovation leads to a significant reduction in memory access for complex weights, specifically by a factor of ${\left(m_{0}\cdot p/m_{1}\right)}^{2}$. However, with the enlargement of the input dimensions from *m*_0_ to *m*_1_, the proposed method incurs a higher count of complex multiplication operations. Despite this, the additional computational cycles required for these operations are minimal, constituting only $\left({\left(m_{1}/m_{0}\right)}^{2}-1\right)$% of those needed for the original FFT-based convolution.

Compared to the method shown in Figure 2b that moves the padding to the frequency domain, the proposed approach substantially diminishes the excess of complex multiplication operations that would otherwise exceed *k*^2^ times by employing interpolation within the frequency domain. Although there is a modest increase in memory access for complex weights, the increment is calculated as ${\left(m_{1}/\left(p\cdot k\right)\right)}^{2}$. This strategic optimization, while curtailing computational complexity, maintains an improved efficiency in memory access, culminating in a marked enhancement in overall system performance.

## 3. Experiment and Analysis

### 3.1. Experimental Data and Platform

#### 3.1.1. Dataset Utilization for Validation

OpenSARShip [20], the premier public Synthetic Aperture Radar (SAR) dataset for ship recognition, was employed to substantiate the performance of the algorithms introduced in this study. This dataset encompasses a collection of 11,346 low-resolution ship images obtained from the Sentinel-1 satellite, with a spatial resolution estimated at 20 m by 22 m. Each imagery within the dataset portrays a solitary ship, with the categorical labels determined by integrating data from the Automatic Identification System (AIS) and applying spatiotemporal interpolation techniques. OpenSARShip presents imagery in two distinct formats: Ground Range Detected (GRD) and Single Look Complex (SLC), both of which accommodate the VV and VH polarizations, thereby providing a robust foundation for the analysis and assessment of SAR-based ship detection and recognition methodologies.

Although the original OpenSARShip dataset offers samples from nearly twenty types of vessels, the scarcity of samples in many categories is insufficient to ensure effective network training and learning. Consequently, in this study, we initially selected four relatively well-represented major categories from the OpenSARShip dataset as our experimental subset: Cargo ships, Tankers, Tugs, and Other types of vessels. This selection is motivated by the fact that these four types of vessels account for approximately 80% of the international shipping trade market, thereby providing a more comprehensive representation. While antecedent studies have indicated minimal influence on recognition accuracy from polarization and imaging modalities [21], this inquiry identified certain limitations within the original dataset. The non-uniformity of image dimensions posed a challenge for deep learning applications, and a marked imbalance existed in the distribution of samples across ship types, with the Cargo class alone accounting for nearly 20,000 instances, vastly outnumbering the other classes.

To counter these issues, a preprocessing phase was undertaken to standardize the image dimensions to a uniform 128 × 128 pixels for the selected ship classes. Furthermore, to mitigate the adverse effects of class imbalance on network training, augmentation of minority class samples is achieved through image rotation. The rotation angle is determined by taking the integer part of the ratio between 360° and the multiple by which the sample needs to be expanded. For instance, if the original dataset contains only 100 bulk cargo ship samples and needs to be expanded to 1000 samples, the images should be rotated 10 times, with each rotation being 36°. Subsequently, the augmented dataset, which has achieved a basic class balance, is randomly divided into training and testing sets in a 7:3 ratio.

Streamlining the network training regimen, all image samples underwent conversion to 8-bit grayscale, a departure from the initial 16-bit data retention policy. This curated subset of the dataset, encompassing the four ship classes, has been designated as OpenSARShip-4 for the purposes of this study. Table 2 delineates the distribution of training and testing samples within the OpenSARShip-4 dataset, tailored for a four-class SAR image ship target classification task. Figure 6 provides a visual representation of the diverse ship class samples encompassed within the OpenSARShip-4 dataset, illustrating the breadth and variety of the curated image samples.

#### 3.1.2. Training and Testing Environment Specifications

The deep neural network models were developed and trained on a high-performance computing setup featuring an Intel Core (TM) i7-8750H CPU (Intel, Santa Clara, CA, USA) operating at 2.20 GHz, complemented by a robust 24 GB GPU memory provided by an NVIDIA GeForce RTX 2070 (Nvidia, Santa Clara, CA, USA) with Max-Q Design and a substantial 32 GB of system memory, all under the Windows 10 operating system. For the evaluation phase, an NVIDIA Jetson Nano 4GB (Nvidia, Santa Clara, CA, USA) development board was deployed, equipped with a powerful quad-core ARM Cortex-A57 MPCore processor (ARM Holdings, Cambridge, UK) and an NVIDIA Maxwell architecture GPU, boasting 128 CUDA^®^ cores and offering a formidable 472 GFlops of computational throughput.

#### 3.1.3. Training Protocol

The training regimen incorporated a batch size of 32, with optimization managed by the Ranger optimizer and regulated by the LambdaLR learning rate scheduler. A cosine annealing learning rate strategy was adopted to refine the training dynamics. The commencement of the training phase involved a warm-up period spanning the first 20 epochs, intended to gradually stabilize the learning process. This was followed by an extended training period, culminating in a total of 100 epochs. The model configuration that demonstrated the most exceptional performance metrics at the culmination of the training sessions was selected and preserved as the definitive architectural and parameterization for the final model.

### 3.2. Input Block Decomposition Result Analysis

The current section details an experimental investigation designed to validate the performance enhancement capabilities of the input block decomposition technique when applied to Convolutional Neural Network (CNN) models. We executed a series of training and testing experiments on the OpenSARShip-4 dataset, applying three distinct convolutional methods as specified in Table 1. For our experiments, we selected a quartet of Convolutional Neural Networks: the mature and widely recognized AlexNet [22] and ResNet18 [23], alongside the state-of-the-art lightweight designs, MobileOne [24] and GhostNetV3 [25]. The input imagery was uniformly resized to a dimension of 128 × 128 pixels for the testing phase. The precision of the network models was evaluated with single-precision accuracy metrics, and the inference velocity was measured in milliseconds (ms). To ensure the robustness of our findings, each convolutional experiment was conducted 20 times, with the average results being reported. A detailed presentation of the experimental data is provided in Table 3, and an integrated visualization of the outcomes is offered in Figure 7, showcasing the performance improvements afforded by the input block decomposition technique across various CNN models.

The data in Table 3 indicate that, compared to the baselines based on SpaceConv and FFTConv, SplitConv achieves comparable recognition accuracy during training iterations while exhibiting the shortest inference time required. Compared with the SpaceConv baseline, the recognition accuracy of SplitConv exhibited a minor decline of 0.07% for AlexNet and 0.52% for MobileOne. Conversely, ResNet18 and GhostNetV3 demonstrated a modest improvement, with gains of 0.08% and 0.59%, respectively. Concurrently, the inference time was significantly reduced, with AlexNet, ResNet18, MobileOne, and GhostNetV3 experiencing reductions of 153.63 ms, 100.33 ms, 35.92 ms, and 32.44 ms, respectively.

In comparison to the FFTConv baseline, SplitConv sustained nearly equivalent accuracy levels. AlexNet and MobileOne recorded slight decreases of 0.12% and 0.33%, while ResNet18 and GhostNetV3 achieved improvements of 0.13% and 0.78%, respectively. Additionally, the inference time for these four network architectures was further decreased by 32.88 ms, 11.46 ms, 10.93 ms, and 8.58 ms, respectively.

These results underscore the nuanced impact of SplitConv on the performance of various neural network architectures, highlighting both the potential for efficiency gains and the maintenance of competitive accuracy levels.

Continuing our inquiry, we proceed to examine how variations in the input image size affect the efficacy of our proposed method. Prior empirical data have corroborated the compatibility of our technique with both network architectures in question. Nonetheless, in light of space constraints, the current experimental series is centered around the widely adopted Convolutional Neural Network, ResNet18, as the benchmark. The input dimensions were systematically increased from 4 × 4 to 256 × 256 pixels, each size being tested in a square aspect ratio. To ensure the reliability of our findings, each configuration was executed 20 times, with the mean outcome being reported.

Acknowledging the established equivalence in accuracy trajectories among the three convolutional methodologies from our prior investigation, this study concentrates on elucidating the relationship between input size and inference time. Figure 8 provides a comparative depiction of the inference time acceleration for FFTConv and SplitConv in juxtaposition to SpaceConv, as the dimensions of the input images are progressively scaled up. This graphical representation quantifies the performance gains of the FFT-based convolutional methods, particularly as the image size grows, offering valuable insights into the scalability and computational efficiency of our proposed approach.

Figure 8 elucidates a consistent trend where FFTConv and SplitConv consistently outperform SpaceConv across all input array sizes examined. With diminutive input arrays (4 × 4, 8 × 8), SplitConv exhibits slightly lower inference velocities relative to FFTConv. However, as the dimensions of the input escalate, especially at the thresholds of 64 × 64 and 128 × 128, SplitConv demonstrates a marked increase in inference speed, ultimately outpacing FFTConv. This observation suggests that while FFTConv may offer greater efficiency for smaller input sizes, its relative performance advantage attenuates with the enlargement of input dimensions. Conversely, SplitConv appears to excel in scenarios involving larger input arrays, indicating a propensity for enhanced scalability when processing voluminous data sets. These findings underscore the methodological adaptability of SplitConv, which is particularly beneficial in applications requiring the manipulation of extensive data arrays.

### 3.3. Composite Zero-Padding Result Analysis

This section presents an experimental investigation aimed at ascertaining the performance enhancement capabilities of the composite zero-padding technique when integrated into Convolutional Neural Network (CNN) models. We conducted a comparative analysis of three distinct zero-padding methodologies—termed “Original”, “FFT”, and “Composite”—applied to the OpenSARShip-4 dataset, as previously discussed in Section 2.4. The FFT-based convolutional approach eschews the utilization of shareable complex weights, opting instead for a frequency-domain computation following the FFT transformation of each convolutional layer. An increased *p*-value facilitates a larger pool of shareable weights but may introduce a trade-off concerning accuracy. For the experiments within this study, *p* is defined as 2.

Two well-established CNN architectures, AlexNet and ResNet18, are engaged for the experimental trials, with the input imagery uniformly resized to 128 × 128 pixels. The fidelity of the network models is evaluated using single-precision accuracy metrics, while the inference velocity is quantified in milliseconds. To ensure the reliability of our findings, each iteration of the “composite zero-padding” experiment is replicated 20 times, with the mean outcome being reported. A detailed presentation of the experimental data is provided in Table 4, and an integrated visualization of the results is offered in Figure 9, showcasing the comparative performance of the three zero-padding techniques across various metrics of interest.

An analysis of the data presented in Table 4 reveals that the composite method not only maintains comparable recognition accuracy throughout the training process but also outperforms the original and FFT methods in terms of inference speed. Specifically, when compared to the original method, there is a slight reduction in recognition accuracy for AlexNet and ResNet18 by 0.06% and 0.17%, respectively. In contrast, MobileOne and GhostNetV3 show a respective increase in accuracy by 0.46% and 0.31%. Concurrently, the inference time for these four network architectures is notably decreased, with reductions of 155.89 ms, 102.47 ms, 32.75 ms, and 30.91 ms, respectively.

Furthermore, when juxtaposed with the FFT method, the composite method’s accuracy remains nearly identical. There is a modest increase in accuracy for AlexNet and GhostNetV3 by 0.13% and 0.15%, respectively, while ResNet18 and MobileOne exhibit a minor decrease in accuracy by 0.20% and 0.23%, respectively. In addition, the inference time for these networks is further reduced, with savings of 30.29 ms, 10.62 ms, 10.74 ms, and 4.67 ms, respectively.

These findings underscore the balanced performance of the composite method, which offers a competitive edge in both recognition accuracy and inference efficiency, setting a new benchmark for comparison against existing methods’ precision.

### 3.4. Split_ Composite Result Analysis

In this section, we present experimental validation of the performance enhancement capabilities of the Split_ Composite method introduced in this paper for Convolutional Neural Network models.

To assess the performance of the model, we employ accuracy and inference time as the metrics for ship recognition evaluation. Accuracy serves as a measure of the model’s classification precision, applicable to classification problems, including the identification of ships. It is defined as the ratio of the number of correct predictions to the total number of predictions made. Inference time, on the other hand, refers to the duration required for the model to perform inference on a single input sample, encompassing the processes of classification or detection. This metric is crucial for gauging the model’s real-time capabilities in practical applications.

We compare the performance of the Split_ Composite method with that of the Winograd [26] and TensorRT [27] convolution optimizations on the OpenSARShip-4 dataset, utilizing the hardware configuration detailed in Section 3.1.2 for training and testing CNN experiments. The Winograd convolution, akin to its FFT counterpart, is a well-established approach for accelerating convolution operations, while TensorRT, NVIDIA’s deep learning inference engine, is tailored for the rapid deployment of GPU-accelerated deep learning models. For the experiments, we engaged two prevalent CNN architectures, AlexNet and ResNet18, with all tests standardized to an input image size of 128 × 128 pixels. The fidelity of the network models was gauged using single-precision accuracy metrics, and the inference velocity was quantified in milliseconds. To ensure the reliability of our findings, each experimental condition was iterated 20 times, with the mean result being reported. A detailed presentation of the experimental data is encapsulated in Table 5, with a synthesized overview of the results graphically represented in Figure 10.

Upon reviewing the data in Table 5 and examining the trends depicted in Figure 10, it becomes evident that the Split_ Composite method surpasses the baselines set by Winograd and TensorRT not only in terms of overall training efficiency but also in achieving the fastest inference times. Specifically, when compared to the Winograd method, the Split_ Composite method enhances the recognition accuracy of MobileOne by 0.85%, while it slightly reduces the accuracy of AlexNet, ResNet18, and GhostNetV3 by 0.17%, 0.03%, and 0.19%, respectively. Concurrently, there is a significant reduction in inference time, with AlexNet, ResNet18, MobileOne, and GhostNetV3 experiencing decreases of 53.63 ms, 38.48 ms, 16.81 ms, and 13.95 ms, respectively.

In contrast to the TensorRT method, the Split_ Composite method demonstrates an overall improvement in recognition accuracy, particularly for MobileOne and GhostNetV3, with increases of 0.85% and 0.56%, respectively. For AlexNet and ResNet18, the accuracy is marginally improved by 0.05% and 0.15%, respectively. Additionally, the inference times for these four networks are further reduced, with AlexNet and ResNet18 showing reductions of 11.87 ms and 11.53 ms and MobileOne and GhostNetV3 by 7.12 ms and 6.83 ms, respectively.

From a comprehensive standpoint, the Split_ Composite method ensures the preservation of accuracy trends during iterative training while concurrently reducing computational complexity and memory access requirements. These optimizations culminate in an improved inference velocity, showcasing the effectiveness of our approach in augmenting the operational efficiency of CNN models without compromising recognition precision.

## 4. Discussion

### 4.1. Practical Application

Our method increases the speed of inference, enabling rapid analysis of SAR images, which is essential for real-time maritime surveillance. The increased speed of reasoning allows for fast and accurate identification of vessels, which is critical for instant decisions to track and identify threats at sea or detect illegal activity. While increasing the speed of reasoning, our approach maintains a high level of identification accuracy, guaranteeing that surveillance systems can reliably identify and classify various types of maritime vessels, which is critical for applications such as search and rescue missions, as accurate identification of vessels can greatly improve the efficiency of mission execution.

The efficiency of our approach in terms of memory access and computing requirements makes it suitable for deployment on resource-constrained platforms, such as mobile or embedded systems used in maritime patrol units or unmanned aerial vehicles (UAVs). By optimizing the performance of CNNs for SAR image analysis, our approach may reduce the need for more expensive hardware and make high-quality maritime surveillance more accessible to a wider range of organizations. In addition, advances in SAR image processing are likely to spur further research and development in the field of maritime surveillance, potentially leading to more innovation building on the success of our Split_ Composite approach.

### 4.2. Model Complexity and Interpretability

Although the Split_ Composite method has demonstrated significant advancements in enhancing the computational efficiency of the algorithm, these improvements have concurrently introduced an escalation in the model’s internal complexity. This increased complexity primarily stems from the multi-tiered weight sharing and sophisticated zero-padding strategies embedded within the model’s architecture. While these methodologies significantly alleviate the computational load, they concurrently augment the obscurity of the model’s inferential mechanisms, thereby complicating the elucidation of the rationale behind specific predictions. Furthermore, to expedite the convolution process via FFT, it is imperative to meticulously calibrate several pivotal parameters, including the zero-padding factor and the partition size for block decomposition. The optimization of these parameters is not guided by intuitive principles, which exacerbates the intricacy of model calibration and escalates the complexity of model stewardship. This is particularly pertinent in domains demanding a high level of interpretability, such as in military sectors, where the current models’ complexity may fall short of the rigorous standards for transparency in the decision-making processes.

Confronted with these challenges, forthcoming research endeavors will concentrate on the innovation of sophisticated techniques for model interpretability designed to demystify the underlying decision-making processes of these models. Efforts will also be directed toward streamlining the model architecture, mitigating the intricacies of parameter optimization, and enhancing the model’s transparency by refining the strategy for parameter adjustment. These initiatives are intended to bolster user confidence in the model and to ensure that our model is not only computationally expedient but also robust, comprehensible, and verifiable for practical applications in SAR maritime surveillance.

## 5. Conclusions

This manuscript presents an innovative convolution acceleration method, Split_ Composite, designed to mitigate the inefficiencies associated with frequent transformation operations in the frequency-domain convolution for SAR image ship target recognition tasks. By amalgamating input block decomposition with a composite zero-padding strategy, our approach markedly augments the processing efficiency and performance of Convolutional Neural Networks (CNNs) when applied to SAR ship imagery. The input block decomposition technique adeptly diminishes the demand for memory bandwidth and lessens computational intricacy through image segmentation and frequency-domain convolution. Meanwhile, the composite zero-padding strategy, capitalizing on the FFT’s inherent periodicity, refines weight-sharing mechanisms, curtailing memory access and arithmetic computational overhead.

Our experimental findings on the OpenSARShip-4 dataset reveal that SplitConv, our proposed method, achieves a substantial enhancement in inference velocity comparable to SpaceConv and FFTConv without compromising recognition accuracy. Moreover, the composite zero-padding technique distinguishes itself by significantly curtailing inference time while sustaining recognition precision on par with the original and FFT approaches. Further experimental scrutiny underscores the enhanced scalability and efficiency of the Split_ Composite method, particularly as the input image dimensions expand. In the realm of large-scale data handling, our method demonstrates superior performance. When juxtaposed with prevalent convolution optimization techniques such as Winograd and TensorRT, Split_ Composite secures a distinct advantage in terms of inference speed while upholding an acceptable accuracy. This holds significant potential for the real-time identification of maritime vessel targets in SAR applications.

## Figures and Tables

**Figure 1 sensors-24-04476-f001:**
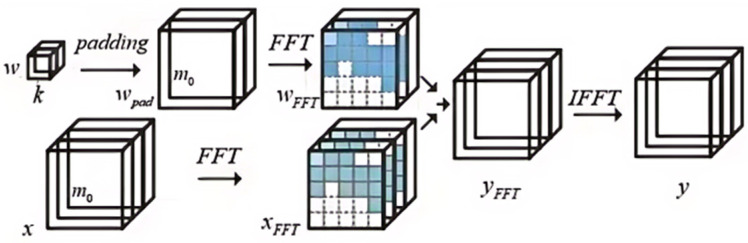
Flow chart of FFT transformation.

**Figure 2 sensors-24-04476-f002:**
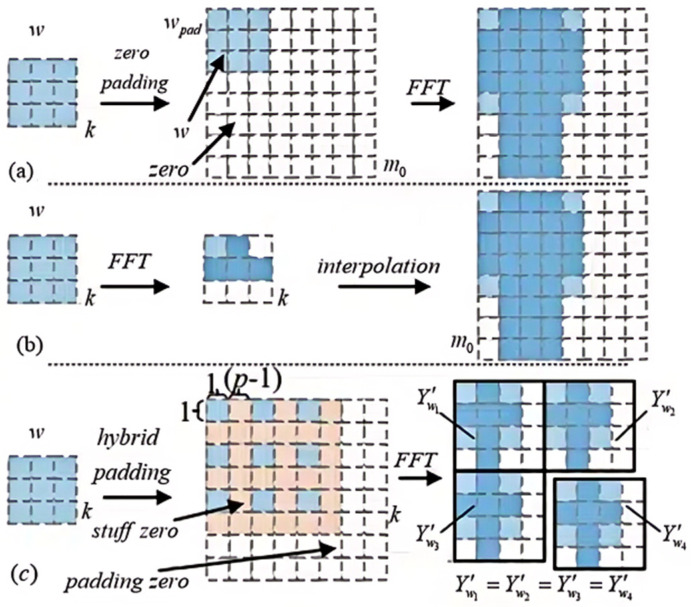
Illustrative examples of zero-padding techniques for weight matrix *w*. (**a**) Original zero-padding. (**b**) Frequency-domain padding. (**c**) Composite zero-padding, where *p* = 2 in this instance.

**Figure 3 sensors-24-04476-f003:**
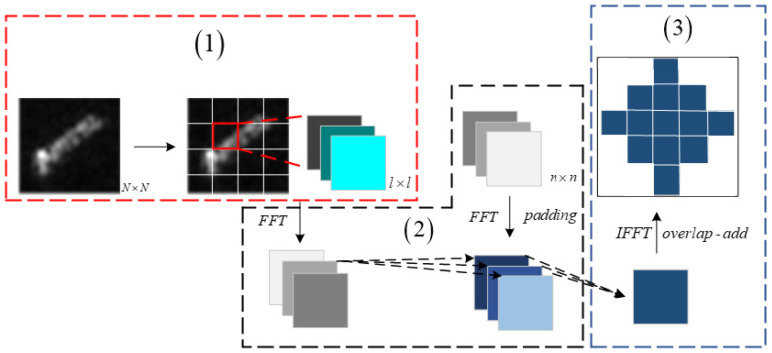
Flow chart of input block decomposition technique.

**Figure 4 sensors-24-04476-f004:**
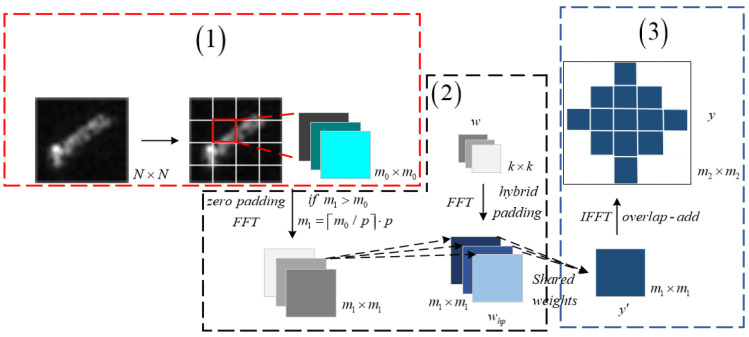
Overall flowchart of the Split_ Composite.

**Figure 5 sensors-24-04476-f005:**
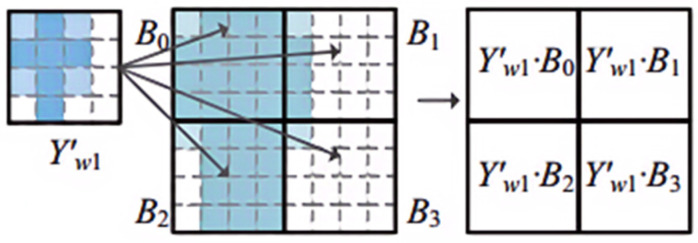
Illustrates the convolution based on Figure 2c. *B*_0_, *B*_1_, *B*_2,_ and *B*_3_ are four blocks of identical size.

**Figure 6 sensors-24-04476-f006:**
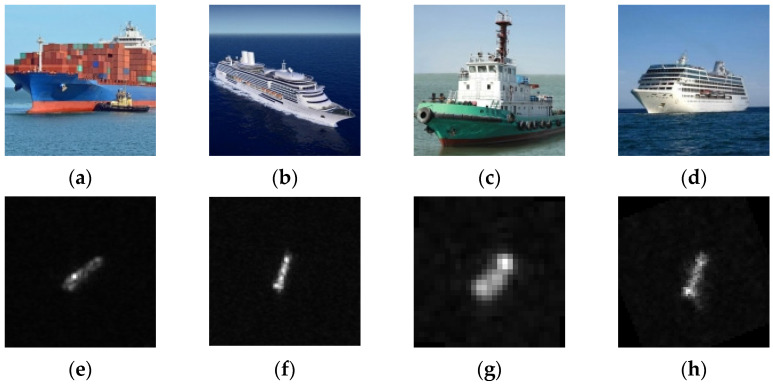
SAR ship samples in OpenSARShip-4 dataset. Where (**a**–**d**) stands for optical images of Cargo, Tanker, Tug, and Other, and (**e**–**h**) stands for SAR images of Cargo, Tanker, Tug, and Other.

**Figure 7 sensors-24-04476-f007:**
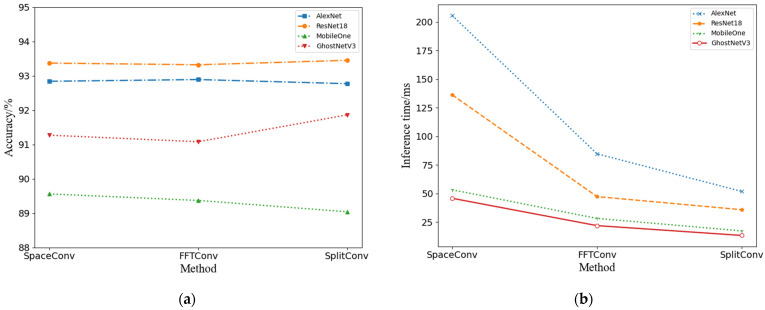
Performance comparison: (**a**) Accuracy comparison and (**b**) Inference time comparison among the four convolution methods.

**Figure 8 sensors-24-04476-f008:**
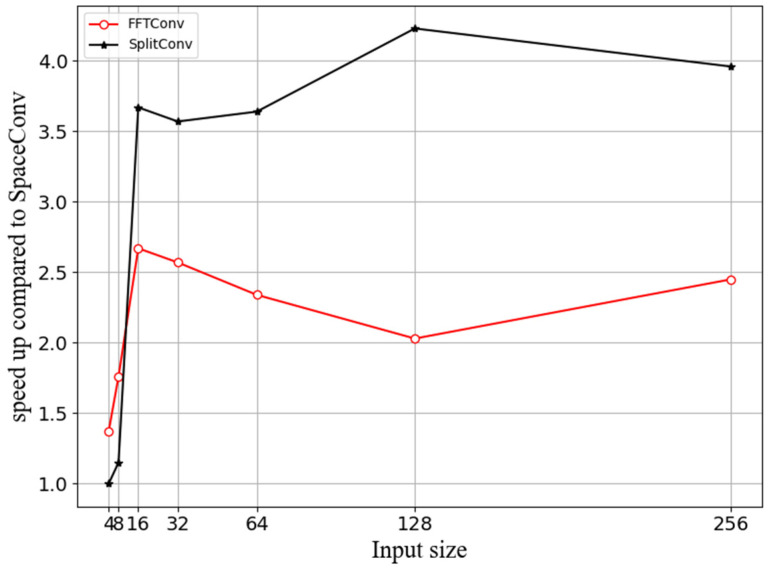
Inference time acceleration coefficient changes between three convolution methods and input sizes.

**Figure 9 sensors-24-04476-f009:**
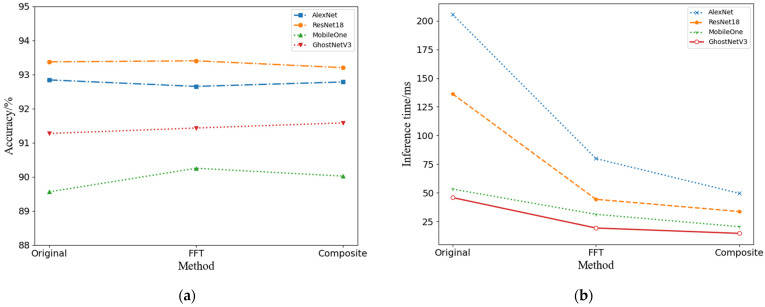
Performance comparison: (**a**) Accuracy comparison and (**b**) Inference time comparison among the four zero-padding methods.

**Figure 10 sensors-24-04476-f010:**
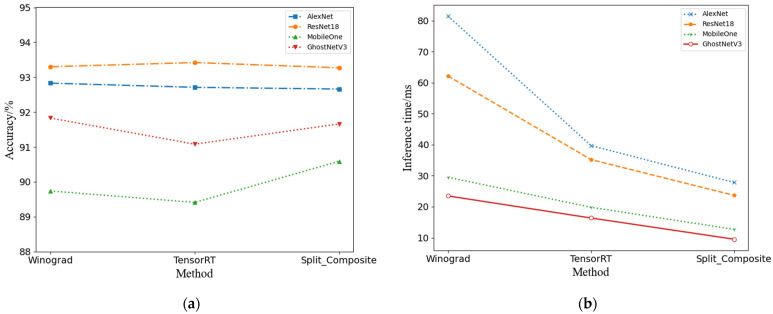
Performance comparison: (**a**) Accuracy comparison and (**b**) Inference time comparison among the four convolution optimization methods.

**Table 1 sensors-24-04476-t001:** Performance rates of networks trained using various convolution methods.

Method	Computational Complexity
SpaceConv	*O(N* ^2^ *n* ^2^ *)*
FFTConv	*O(N* ^2^ *log* _2_ *N)*
SplitConv	*O(N* ^2^ *log* _2_ *n)*

**Table 2 sensors-24-04476-t002:** OpenSARShip-4 sample statistics.

Category	Train	Test	Total
Cargo	1610	690	2300
Tanker	1610	690	2300
Tug	1610	690	2300
Other	1610	690	2300

**Table 3 sensors-24-04476-t003:** Performance comparison of three convolution methods.

Networks	Method	Accuracy/%	Standard Deviations	Inference Time/ms	Standard Deviations
AlexNet	SpaceConv	92.84	0.1069	205.47	80.73
	FFTConv	92.89		84.72	
	SplitConv	92.77		51.84	
ResNet18	SpaceConv	93.37	0.0658	136.21	54.91
	FFTConv	93.32		47.34	
	SplitConv	93.45		35.88	
MobileOne	SpaceConv	89.56	0.2626	53.28	18.41
	FFTConv	89.37		28.29	
	SplitConv	89.04		17.36	
GhostNetV3	SpaceConv	91.27	0.4065	45.85	16.78
	FFTConv	91.08		21.99	
	SplitConv	91.86		13.41	

**Table 4 sensors-24-04476-t004:** Performance comparison of three zero-padding methods.

Networks	Method	Accuracy/%	Standard Deviations	Inference Time/ms	Standard Deviations
AlexNet	Original	92.84	0.4200	205.47	82.6537
	FFT	92.65		79.87	
	Composite	92.78		49.58	
ResNet18	Original	93.37	0.4277	136.21	42.4617
	FFT	93.40		44.36	
	Composite	93.20		33.74	
MobileOne	Original	89.56	0.3527	53.28	14.8736
	FFT	90.25		31.27	
	Composite	90.02		20.53	
GhostNetV3	Original	91.27	0.4367	45.85	13.9733
	FFT	91.43		19.34	
	Composite	91.58		14.67	

**Table 5 sensors-24-04476-t005:** Performance comparison of three convolution optimization methods.

Networks	Method	Accuracy/%	Inference Time/ms
AlexNet	Winograd	92.83	81.47
	TensorRT	92.71	39.71
	Split_ Composite	92.66	27.84
ResNet18	Winograd	93.30	62.16
	TensorRT	93.42	35.21
	Split_ Composite	93.27	23.68
MobileOne	Winograd	89.74	29.52
	TensorRT	89.41	19.83
	Split_ Composite	90.59	12.71
GhostNetV3	Winograd	91.83	23.49
	TensorRT	91.08	16.37
	Split_ Composite	91.64	9.54

## Data Availability

Data can be provided by the corresponding author based upon reasonable request.

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
