# Peer review of "Split_ Composite: A Radar Target Recognition Method on FFT Convolution Acceleration"

_sensors, 2024, doi:10.3390/s24144476_

Round 1

Reviewer 1 Report

Comments and Suggestions for Authors

The authors have introduced an innovative Split_Composite approach that notably enhances inference speed while maintaining high recognition accuracy. This work presents a valuable contribution and demonstrates promising results, hence I recommend it for publication.

A few suggestions for the authors' consideration:

  1. Incorporation of Recent Neural Network Lightweighting Algorithms for On-Orbit Processing: It is noteworthy that there have been numerous advancements in neural network lightweighting algorithms specifically designed for on-orbit (on satellite) processing in recent years. These might offer valuable insights for the authors' future research endeavors, and their incorporation could further optimize the Split_Composite method.

  2. Enhancing Reproducibility: As a public academic publication, reproducibility is paramount. I noticed that the Data Availability Statement is currently empty, and detailed information regarding the datasets used is lacking. If the data is open-source, kindly provide the URLs where they can be accessed. If they are proprietary, please suggest similar open-source datasets and discuss the similarities and differences between your proprietary dataset and the publicly available ones. This will facilitate readers in tracking and reproducing your findings.

  3. Adjustment of Figure Scaling: Figures 7 to 10 appear to have overly small scales, making them difficult to read. It is recommended that the font sizes and scales of these figures be adjusted to standard readability levels.

  4. Improving Article Structure and Flow: Between lines 336 and 228 (assuming this is a typo and should be 336 to 328 or similar, as line numbers typically increase downwards), there seems to be a concentration of lists without sufficient intervening narrative sentences. To enhance readability and assist readers in navigating the paper's framework and structure, it is advisable to integrate more descriptive statements within and around these lists, under appropriate section headings. This will help maintain a smooth flow and improve the overall coherence of the manuscript.

Overall, this is an interesting and impactful study, and with the above-mentioned refinements, it will further strengthen the quality and accessibility of the research

Comments on the Quality of English Language

Acceptable

Reviewer 2 Report

Comments and Suggestions for Authors

Summary

The manuscript presents Split_Composite, a novel method leveraging Fast Fourier Transform (FFT) for convolution acceleration in Convolutional Neural Networks (CNNs) specifically applied to Synthetic Aperture Radar (SAR) ship target recognition. The method aims to reduce memory bandwidth and computational complexity, making it particularly suitable for embedded systems. The study showcases significant improvements in inference speed without compromising recognition precision.

Significance:
The results are interpreted appropriately, showing substantial improvements in computational efficiency. The hypotheses are clearly identified and rigorously tested, leading to significant and well-supported conclusions.

Quality:
The article is well-written and appropriately structured. Data and analyses are presented clearly, adhering to high standards of result presentation.

Scientific Soundness:
The study is correctly designed and technically sound. The methods and tools are described in sufficient detail, ensuring reproducibility. The data is robust enough to draw reliable conclusions.

Interest to the Readers:
The conclusions are relevant and interesting to the readership of the journal. The advancements in SAR technology and CNN acceleration are likely to attract a wide audience.

Overall Merit:
The manuscript offers significant benefits by advancing current knowledge in SAR ship target recognition. The experiments are smartly designed, addressing important challenges in the field.

Figures/Tables:
The figures, tables, and images are appropriate, clearly showing the data and easy to interpret. The data is interpreted consistently throughout the manuscript.

The manuscript presents a well-structured, scientifically sound study that is relevant and interesting to the readership of the Sensors journal. The proposed method, Split_Composite, offers significant advancements in SAR ship target recognition by improving computational efficiency. Based on the strengths outlined and the minor revisions suggested, I recommend the manuscript for publication after minor revisions.

Recommendations for Minor Revisions:

  1. Clarify Methodology: Provide more details on the composite zero-padding method and its integration with Split_Composite to enhance reproducibility.
  2. Expand Discussion: Offer a more detailed discussion on the limitations and potential future research directions.
  3. English Proofreading: Conduct a final round of proofreading by a native English speaker to address any remaining minor language issues.

Reviewer 3 Report

Comments and Suggestions for Authors

The manuscript presents a novel and promising approach to accelerating convolution operations for SAR ship target recognition. However, to enhance its technical rigor and overall clarity, the manuscript would benefit from addressing the outlined technical issues and improving the presentation of experimental results. Additionally, incorporating the suggested references would strengthen the background and justification for your proposed method.

1:The introduction provides a broad overview of SAR and its applications. However, it lacks a precise problem definition that your method aims to solve. Clearly state the specific problem in SAR ship target recognition that Split_Composite addresses.

2:Several claims are made without sufficient references, such as the capabilities of SAR (lines 31-40). Ensure that each significant claim is supported by appropriate citations.

3:The equations presented in the frequency domain convolution section (e.g., equations 1-4) are introduced without sufficient derivation or explanation. Include more detailed derivations to make the methodology clearer.

4: The steps involved in the Split_Composite methodology are described in prose. Including pseudocode or a step-by-step algorithm would enhance clarity and reproducibility (e.g., steps from input block decomposition to FFT transformations).

5:While the method is compared with Winograd and TensorRT, the basis of comparison is not clearly defined. Provide more details on the benchmarking process, including the specific hardware configurations, dataset preprocessing steps, and performance metrics used.

6:The paper mentions data augmentation techniques to balance the dataset (lines 359-362) but lacks details on the specific techniques used. Specify the data augmentation methods and how they impact the model performance.

7:The paper primarily focuses on accuracy and inference time. Including other metrics such as precision, recall, and F1-score, particularly for imbalanced datasets, would provide a more comprehensive evaluation of model performance.

8:The results are presented without statistical analysis. Include statistical tests to demonstrate the significance of the improvements observed with Split_Composite over other methods.

9:Some figures, such as Figure 3 (illustrating zero-padding techniques), are complex and may be difficult to interpret. Simplify these figures or break them into smaller, more focused illustrations.

10:Tables 3 and 4 present performance comparisons but lack standard deviations or confidence intervals. Include these to provide a better understanding of the variability in performance.

11:The discussion section should delve deeper into the implications of the results. Discuss how the improvements in inference speed and accuracy can impact real-world applications of SAR in maritime surveillance.

12:Suggested References:

—"An advanced scheme for range ambiguity suppression of spaceborne SAR based on blind source separation":

This paper provides advanced techniques for improving SAR imaging, which could complement the proposed Split_Composite method, especially in enhancing image quality and target recognition.

—"A survey on deep-learning-based real-time SAR ship detection":

This survey covers various deep learning approaches for SAR ship detection. It would provide a comprehensive background and context for your work, highlighting where Split_Composite fits within the current state of the art.

Comments on the Quality of English Language

It is recommended that the article be submitted to a native English speaker or professional editing service for polishing. This will ensure that the language and grammar are of the highest quality, thereby enhancing the readability and professionalism of the manuscript.
